# Rational Design of Hydrogels for Cationic Antimicrobial Peptide Delivery: A Molecular Modeling Approach

**DOI:** 10.3390/pharmaceutics15020474

**Published:** 2023-01-31

**Authors:** Alfredo Pereira, Elizabeth Valdés-Muñoz, Adolfo Marican, Gustavo Cabrera-Barjas, Sekar Vijayakumar, Oscar Valdés, Diana Rafael, Fernanda Andrade, Paulina Abaca, Daniel Bustos, Esteban F. Durán-Lara

**Affiliations:** 1Departamento de Química Orgánica y Fisicoquímica, Facultad de Ciencias Químicas y Farmacéuticas, Universidad de Chile, Santiago 8380544, Chile; 2Doctorado en Biotecnología Traslacional, Facultad de Ciencias Agrarias y Forestales, Escuela de Ingeniería en Biotecnología, Universidad Católica del Maule, Talca 3480094, Chile; 3Instituto de Química de Recursos Naturales, Universidad de Talca, Talca 3460000, Chile; 4Bio & Nano Materials Lab, Drug Delivery and Controlled Release, Departamento de Microbiología, Facultad de Ciencias de la Salud, Universidad de Talca, Talca 3460000, Chile; 5Center for Nanomedicine, Diagnostic & Drug Development (ND3), Universidad de Talca, Talca 3460000, Chile; 6Unidad de Desarrollo Tecnológico (UDT), Universidad de Concepción, Parque Industrial Coronel, Coronel 3349001, Chile; 7Marine College, Shandong University, Weihai 264209, China; 8Centro de Investigación de Estudios Avanzados del Maule (CIEAM), Vicerrectoría de Investigación y Postgrado Universidad Católica del Maule, Talca 3460000, Chile; 9Drug Delivery & Targeting, Vall d’Hebron Institut de Recerca (VHIR), Universitat Autònoma de Barcelona (UAB), 08035 Barcelona, Spain; 10Centro de Investigación Biomédica en Red de Bioingeniería, Biomateriales y Nanomedicina (CIBER-BBN), Instituto de Salud Carlos III, 28029 Madrid, Spain; 11Functional Validation & Preclinical Research (FVPR)/U20 ICTS Nanbiosis, Vall d’Hebron Institut de Recerca (VHIR), Universitat Autònoma de Barcelona (UAB), 08035 Barcelona, Spain; 12Departament de Farmàcia i Tecnologia Farmacèutica i Fisicoquímica, Facultat de Farmàcia i Ciències de l’Alimentació, Universitat de Barcelona (UB), 08028 Barcelona, Spain; 13Laboratorio de Bioinformática y Química Computacional, Departamento de Medicina Traslacional, Facultad de Medicina, Universidad Católica del Maule, Talca 3480094, Chile

**Keywords:** cationic antimicrobial peptides, hydrogels, molecular docking, molecular dynamics simulations, virtual-screening, antibiotic resistance

## Abstract

In light of the growing bacterial resistance to antibiotics and in the absence of the development of new antimicrobial agents, numerous antimicrobial delivery systems over the past decades have been developed with the aim to provide new alternatives to the antimicrobial treatment of infections. However, there are few studies that focus on the development of a rational design that is accurate based on a set of theoretical-computational methods that permit the prediction and the understanding of hydrogels regarding their interaction with cationic antimicrobial peptides (cAMPs) as potential sustained and localized delivery nanoplatforms of cAMP. To this aim, we employed docking and Molecular Dynamics simulations (MDs) that allowed us to propose a rational selection of hydrogel candidates based on the propensity to form intermolecular interactions with two types of cAMPs (MP-L and NCP-3a). For the design of the hydrogels, specific building blocks were considered, named monomers (MN), co-monomers (CM), and cross-linkers (CL). These building blocks were ranked by considering the interaction with two peptides (MP-L and NCP-3a) as receptors. The better proposed hydrogel candidates were composed of MN3-CM7-CL1 and MN4-CM5-CL1 termed HG1 and HG2, respectively. The results obtained by MDs show that the biggest differences between the hydrogels are in the CM, where HG2 has two carboxylic acids that allow the forming of greater amounts of hydrogen bonds (HBs) and salt bridges (SBs) with both cAMPs. Therefore, using theoretical-computational methods allowed for the obtaining of the best virtual hydrogel candidates according to affinity with the specific cAMP. In conclusion, this study showed that HG2 is the better candidate for future in vitro or in vivo experiments due to its possible capacity as a depot system and its potential sustained and localized delivery system of cAMP.

## 1. Introduction

About two million fatalities were attributed to antimicrobial resistance (AMR) in 2020 alone, a number that grew due to the COVID-19 pandemic’s indiscriminate antibiotic usage [1]. By 2050, it is predicted that over 10 million people will have died as a direct result of antibiotic resistance [2]. Nosocomial pathogenic micro-organisms with rising degrees of antibiotic resistance are referred to by the abbreviation ESKAPE for the Gram-positive species *Enterococcus faecium* and *S. aureus* and the Gram-negative *K. pneumoniae*, *A. baumannii*, *P. aeruginosa* and *Enterobacter* species. ESKAPE pathogens tend to inhabit hospital environments, and they are the most common cause of fatal infections in immunocompromised patients [3,4]. Despite a vital need for novel antimicrobial drugs, just a few new viable antibiotics are currently in clinical development. In this context, AMPs appear as promising candidates in the fight against multi-drug resistance bacterial infections due to their broad range of activities and low toxicity [5]. AMPs are small proteins that have antibacterial broad-spectrum and immunomodulatory effects on infections produced by micro-organisms such as bacteria, viruses, fungi, or parasites. These peptides can be characterized according to their sequence-structure composition, e.g., net-charge, secondary and tertiary architecture, solubility, electrostatic components, and source of origin (natural or synthetic) [6]. Unlike antibiotics, which disrupt the synthesis of peptidoglycan in bacterial cell walls or inhibit the DNA gyrase, AMPs operate by creating membrane pores and attaching to molecules that are precursors to membranes [7,8]. One of the most abundant AMPs is cAMP, which can disrupt the bacterial membrane, resulting in cell death via osmotic shock [6]. The remarkable spectrum of functions that cAMP may perform includes antibacterial action against both Gram-positive and Gram-negative bacteria [9]. Furthermore, cAMPs are auspicious candidates to replace conventional antibiotics; however, they possess some drawbacks, such as their short half-life due to enzyme degradation, low bioavailability and solubility, and low activity in physiological conditions, among others [10,11]. An effective action to resolve the problem is to apply drug delivery nano-systems (DDNSs) that allow a sustained or prolonged release of AMPs. Thus, a rational design of effective cAMP-delivery nano-systems that provide protection and an effective release is essential for the successful implementation of cAMP treatments [12]. Some studies have proved the efficacy of combining hydrogels and cAMPs, enhancing the antimicrobial potency compared to the free-cAMP alternative and low cell toxicity [13,14]. Hydrogels have received growing attention in recent years as DDNSs due to their exclusive properties, such as high biocompatibility, tunable release rate, and versatility with regard to being loaded with different molecules. Consequently, to rationally design hydrogels, taking into consideration the interaction affinity with cAMPs is fundamental for achieving high-release performance. For example, unpredictable slow releases of drugs from hydrogels were attributed to some associative interactions between the drug and the matrix, impairing the release rate [15]. Stability and aggregation are other common difficulties regarding DDNS preparation, i.e., when an amphiphilic peptide is encapsulated into a hydrophobic system that includes an aqueous layer, it could leak from the systems [16]. The rational design of the on-demand formulations must be accurate in order to avoid the usual DDNS limitations. Under this new approach, injectable and stimuli-responsive hydrogels seem to be good candidates to solve these concerns. These platforms are highly attractive biomaterials in drug delivery and tissue engineering [17]. This class of hydrogels responds to small variations in environmental conditions, such as light, ionic strength, magnetic fields, pH, and temperature. Among these, the most commonly studied hydrogels are responsive either to pH or temperature [18,19,20]. Until now, a wide variety of stimuli-responsive hydrogels has been reported; among them is the hydrogel of poly(N-isopropylacrylamide) (PNIPAM), the gold standard polymer in the field of thermoresponsive hydrogels (TRH) [21]. PNIPAM has a lower critical solution temperature of 32−34 °C [22,23]. Given its exclusive hydrophilic–hydrophobic transition behavior, various TRH PNIPAM-based hydrogels have been created for the controlled delivery of hydrophilic protein drugs. A special advantage regarding these hydrogels is that the loading of hydrophilic proteins can be achieved in aqueous media at a low temperature, which diminishes the chance of the denaturation of proteins in a hostile environment [18]. By using theoretical-computational methods such as docking and MDs, it is possible to understand and predict the interaction between hydrogels and different bioactive compounds such as AMP [24]. With these analyses, it could be possible to adjust the specific interaction level of hydrogel-cAMP, modifying the hydrogel structure with different building blocks. The above could guarantee the potential use of hydrogels as a depot system and a localized and sustained release platform of cAMPs. Therefore, in this study, we employed a set of docking and MDs that allowed us to propose a rational selection of hydrogel candidates based on the propensity to form intermolecular interactions with specific cAMPs as an innovative approach to the rational design of hydrogels applied to the sustained and localized release of cAMPs.

## 2. Materials and Methods

### 2.1. AMP Selection

After a rigorous search, two cAMPs were selected—MP-L (ID: DRAMP18623), and NCP-3a (ID: DRAMP18636)—as cAMP models from the database http://dramp.cpu-bioinfor.org (accessed on 12 March 2022). The requirements of selection were (i) having a net positive charge, (ii) showing antimicrobial activity against the largest number of targets from the ESKAPE pathogens, and (iii) containing a number equal to or lower than 20 amino acids. The cAMP properties are shown in Table 1. The secondary and tertiary structures of each cAMP were predicted with PEP-FOLD 3.5 server [25]. Both cAMPs were energy minimized by using the conjugate gradient algorithm in a vacuum; then, each cAMP was embedded into a box of SPC (single-point charge) water molecules and chloride ions were added to neutralize the systems. Subsequently, the systems were equilibrated through 25 ns of all-atom MDs in an isobaric-isothermal NPT ensemble at 1 atm and 298.15 K with Maestro/Schrödinger suite [26] and OPLS3e as force fields [27]. The endpoint of cAMP in each simulation was collected for further analysis.

### 2.2. Hydrogel Building Blocks Selection

The selection of the building blocks such as MN, CM, and CL was made with the aim of evaluating the influence of the polarity and non-ionic interactions over the affinity of the hydrogel-peptide complex without negatively affecting the stimulus sensitivity and mechanical properties of the hydrogel. MN, CM, and CL are shown in Figure 1.

The structural units of the hydrogel systems, called “building blocks”, were sketched in Maestro/Schrödinger suite and prepared at pH 7 according to [24].

### 2.3. Analysis of the Affinity between Hydrogel Building Blocks and cAMPs

The building blocks were used as inputs jointly with each cAMP to perform a virtual screening based on docking. For this, two different protocols and docking programs were employed. First, we used the Autodock Vina program [28] to dock each MN, CM, and CL against each cAMP (named Protocol 1). The Autodock Vina makes use of a semi-flexible approach in that it only allows the ligand (building blocks in this case) to be made flexible. In the second protocol (Protocol *2*), the Induced-Fit docking (IFd) approximation of glide software was used [29,30] with the most rigorous algorithm possible—XP (Extra Precision)—and using the OPLS3e force field. In both protocols, the final number of poses per pair (building block-cAMP) was set at 10. Then, all the poses through boxplots comparing the binding energy of each segment against the rest of its group in both protocols were statistically analyzed. The best building block belonging to each group (MN, CM, and CL) based on the binding affinity with cAMP were selected for the next step.

### 2.4. Ensemble and Simulations of Hydrogel-cAMP Complexes

Four systems were ensembled: MN3-CM7-CL1 (termed as HG1) and MN4-CM5-CL1 (termed as HG2) hydrogels with each cAMP (MP-L and NCP-3a) independently. First, the LEAP program from AmberTools21 suite [31] was used to build substructures of hydrogels, considering the proportions of MN, CM and CL used in [24]. The HG1 substructure was constituted by two chains, each of 20 MN3 plus 5 CM7 (randomly distributed), cross-linked by a CL1. The HG2 substructure was constituted by 2 chains, each with 20 MN4 plus 5 CM5 (randomly distributed), cross-linked by a CL1.

Then, the chains were equilibrated by MDs with the same parameters previously employed in the simulations of the cAMPs alone. The endpoint of each simulation was collected as input to build the hydrogel-cAMP complex. For this, Packmol software [32] was used to build a sphere of 60 Å of the radius where four hydrogel chains were placed, and another inner sphere of 50 Å of the radius where 10 cAMP molecules were positioned. Subsequently, the four systems were minimized, solvated, and neutralized with NaCl counter ions; finally, each system was simulated by 50 ns in the same condition as the previously described simulations. The MDs allow us to identify the main intermolecular interactions and forces between the hydrogels and cAMPs. All the analyses were performed over the entire simulation time and consisted of the gyration radius of each hydrogel, the secondary structure fluctuation of both peptides, the intermolecular contacts and non-bonding energies (van der Waals, electrostatic and total), and the interactions (HBs and SBs) between the hydrogels and the cAMPs. The intermolecular contacts were computed with an in-house TCL script and run in VMD software [33]. Regarding the non-bonding energies and interactions, these were calculated from the Maestro/Schrödinger suite and its Simulation Event Analysis tools. All the plots and statistical analysis were generated with R v.4.2.2 and RStudio v. 2022.07.0.

## 3. Results and Discussion

### 3.1. Ranking of Building Blocks by Molecular Docking

Figure 2, Figure 3 and Figure 4 present the building block rankings using the protocols based on the Autodock Vina (protocol 1) and IFd approximation of glide (protocol 2). The rankings of the MNs (Figure 2), CMs (Figure 3), and CLs (Figure 4) are shown considering the peptides MP-L and NCP-3a as receptors. For both strategies, the statistical comparison between building blocks were carried out by taking the 10 best poses of each docking run. Then, the selection of the best candidates was made based on the median value of the binding energy in both protocols individually. In both cases, the lower the value, the better the candidate.

Overall, the results showed that the best candidates obtained coincide with the peptides studied; however, important differences were observed between the proposed strategies. For MN ranking, MN3 was the best candidate for both peptides using protocol 1, whereas MN4 was the best candidate for both peptides using protocol 2. For the CM ranking, using protocol 1, the best candidate for both peptides was CM7; however, in the case of protocol 2, the best candidate for both peptides was CM5. Finally, in the CL ranking, for both strategies and considering the two peptides studied, the best candidate turned out to be CL1. It is important to mention that when protocol 2 was used, CL4 also appeared as a good candidate, showing no significant difference with CL1; however, the median score value was still lower in CL1. The variations observed in the building blocks when comparing both protocols may be attributed to the specific docking method applied in each protocol. As mentioned above, AutoDock uses a semi-flexible variant; in contrast, IFd is based on the Induced-Fit hypothesis, which states that when a receptor (peptides) binds to a ligand (building blocks), it experiences conformational modifications, allowing for a better fit between the two molecules. The IFd method is thought to be more accurate than typical rigid-body docking methods because it takes into consideration the dynamic nature of the binding. In practice, the binding energy values obtained with both protocols are not comparable because they are obtained from different scoring functions. Therefore, it was opted to select the best candidates in each protocol for the following steps. So, considering both strategies, the proposed hydrogel candidates were formed by the following blocks: MN3-CM7-CL1 (protocol 1) and MN4-CM5-CL1 (protocol 2). For the next analysis, these hydrogels will be named HG1 (derived from protocol 1) and HG2 (derived from protocol 2).

### 3.2. Molecular Dynamics Simulations

In order to study the behavior of hydrogels and their interactions with cAMPs over time and in an environment that mimics reality, MDs were performed. Four systems were built, which were the combination of the two peptides MP-L (P1) and NCP-3a (P2) and the two proposed hydrogels (HG1 and HG2). It is worth mentioning that for a better understanding, the systems were labeled as follows: HG1_P1, HG1_P2, HG2_P1, and HG2_P2.

First, the structural stability of the hydrogels and cAMP in the systems proposed was analyzed. Figure 5A shows the radius of gyration of the hydrogels during the 50 ns of simulation. Here, it can be observed that both HGs have similar behavior and that the radius of gyration does not vary too much between them, reaching values close to 40 Å. However, statistically significant differences are observed (Figure 5B). Both HGs have a larger radius of gyration in the presence of P1, and a smaller radius of gyration when in the presence of P2. This allows us to infer that small structural variations of the HGs occur to improve the interactions with each type of cAMP. Regarding the stability of the cAMP, Figure 6 shows the secondary structure elements (SSE) of each one of the cAMP’s residues in the four proposed systems. It is worth mentioning that each system (simulated in triplicate) was composed of 10 cAMPs, so each graph shows the statistical mode of the secondary structure for each residue at each specific time point. Overall, the results obtained show that the cAMPs maintain their secondary structure during the simulation. Figure 6A,B shows that the cAMP P1 retains its alpha-helical structure, and only small changes are observed between residues 12 to 15. In the case of the cAMP P2 (Figure 6C,D), it occurs in a similar way, where both beta sheets and the turn region in the center of the structure are maintained. From this last result, it can be inferred that both cAMPs do not lose their secondary structure when interacting with the HGs, which is an important advantage for the cAMPs’ stability to maintain their antimicrobial function.

The following analysis describes and quantifies the interactions that occurred between the HGs and cAMPs. For this, firstly, the cAMPs that were less than 3 Å from the HGs were counted (the results are shown in Figure 7). It can be seen that in the four systems the interaction between the 10 peptides and the HGs is achieved quickly; however, not all peptides remained within 3 Å during simulation. Statistically, it is also possible to observe differences between the systems, wherein the most notable is that the median value of the cAMPs is higher for both HGs when they interact with P2 and lower for both HGs when they interact with P1. Subsequently, the interactions between the HGs and cAMPs were identified. Due to the chemical nature of the selected blocks and the amino acids that form the cAMPs, the HBs and SBs were quantified. Figure 8A shows the HBs formed by the MN of the HGs and the cAMPs. Overall, it can be seen that the number of HBs formed is low; according to the number of cAMPs in each system, where the systems HG1_P1, HG2_P1, and HG2_P2 barely reach five bonds, the only system that achieves a greater amount (approximately 10 bonds) is HG1_P2, which, to clarify, has the MN3. Figure 8B shows the HBs formed between the CM of the HGs and the cAMPs. Here, the number of HBs increases drastically, reaching values of over 20 HBs in each system. The most noticeable is the great difference between the HG1 and HG2 hydrogels, where HG2 practically doubles the number of HBs that HG1 can form with the cAMPs. This difference is due to the chemical structure of the CM that forms the hydrogel, since CM7 (HG1 hydrogel) has only one carboxylic acid in its structure, while CM5 (HG2 hydrogel) has two carboxylic acids in its structure. Another type of interaction formed between hydrogels and peptides are SBs, which by definition are HBs that occur between a positively charged donor and a negatively charged acceptor. In our analysis, a similar amount of SBs (Figure 8C) and HBs (Figure 8B) were detected; in addition, the trend in which the largest number of SB is formed by the HG2 hydrogel is maintained. This is because both interactions are formed by cationic amino acids from the peptides and deprotonated carboxylic acids from the hydrogels (Figure 9). It is worth mentioning that given the chemical nature, no interaction by SBs was found in the MN, nor HBs and SBs in the CL blocks.

Finally, to complement the previous results, the electrostatic (elec), van der Waals (vdW) and non-bonding (elec + vdW) interaction energies between hydrogels and cAMPs were calculated. For this analysis, the lower energy, the higher the affinity between hydrogels and cAMPs. The results obtained show evident differences between hydrogels, where clearly the hydrogel HG2 has drastically lower electrostatic energies than HG1 (Figure 10A); on the other hand, the hydrogel HG1 has lower van der Waals energies than HG2 (Figure 10B). These differences are directly related to the previous results and, in turn, are explained by the aforementioned chemical nature of the CM—that is, the greater amount of deprotonated carboxylic acids involved in the formation of HBs and SBs. Furthermore, it is important to mention that the vdW energies are associated with hydrophobic interactions, which, in this case, should be occurring between the free isopropyl group of CM7 and different aliphatic regions of the peptides. However, these vdW energies become insignificant when compared to electrostatic energies, since if we analyze the non-bonding total energy (Figure 10C) it is mainly governed by electrostatic energy.

## 4. Conclusions

Two docking strategies were used to select the best building block for designing hydrogels for antimicrobial peptide localized delivery. Both strategies selected different MN and CM, but the same CL. The proposed hydrogel candidates were composed of MN3-CM7-CL1 (HG1) and MN4-CM5-CL1 (HG2) for both cAMPs studied. HG2 has greater interactions with the studied peptides due to its two carboxylic acids present in its structure, as shown by MD results which indicate greater HBs and SBs formed by HG2. Both hydrogels rapidly interact with the peptides without altering their structural stability, making HG2 a better candidate for future in vitro or in vivo experiments.

The importance of hydrogels as drug delivery systems is unquestionable; their selection and optimization can be effectively achieved with theoretical methods [12,24], as is reported by several studies such as the study of Changying et al. [34], where through experimental validation they were able to confirm a computational pipeline similar to the one reported here and thereby identify telodendrimer nanocarriers for specific drugs. The obtained results demonstrate that it could be possible to adjust “on demand” the specific interaction level of hydrogel-cAMP by modifying the hydrogel structure with different building blocks that consider the cAMP structure. The above could guarantee the potential use of hydrogel as a depot system and a localized and sustained release platform for cAMPs.

Finally, we propose a quick-to-implement strategy for the rational selection of building blocks for the formation of hydrogels with applications in different areas such as pharmaceutics, nanomedicine, and biotechnology.

## Figures and Tables

**Figure 1 pharmaceutics-15-00474-f001:**
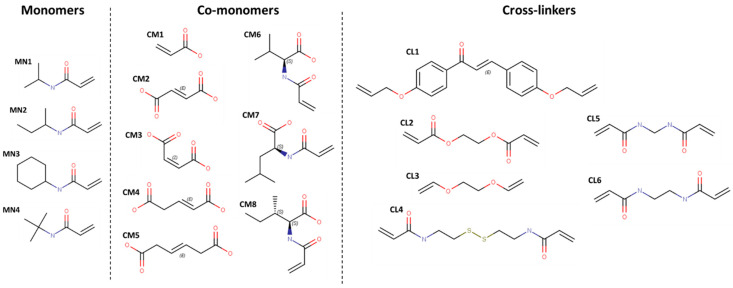
Chemical structures of MN, CM and CL.

**Figure 2 pharmaceutics-15-00474-f002:**
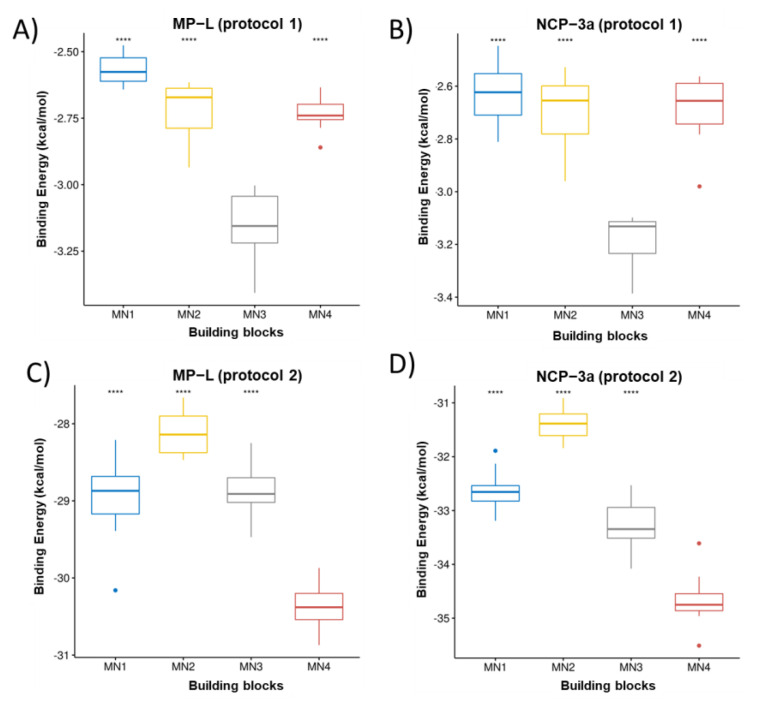
Ranking of MN. The boxplots of the cAMP-MNs binding energies calculated from the VINA scoring function are shown in (**A**,**B**) for the peptides MP-L and NCP-3a, respectively. The same as (**A**,**B**) but for the binding energy obtained from the glide scoring function (IFd) are shown in (**C**,**D**). **** correspond to *p* < 0.0001 of comparative statistical difference (using the *t*-test) with the lowest energy building block.

**Figure 3 pharmaceutics-15-00474-f003:**
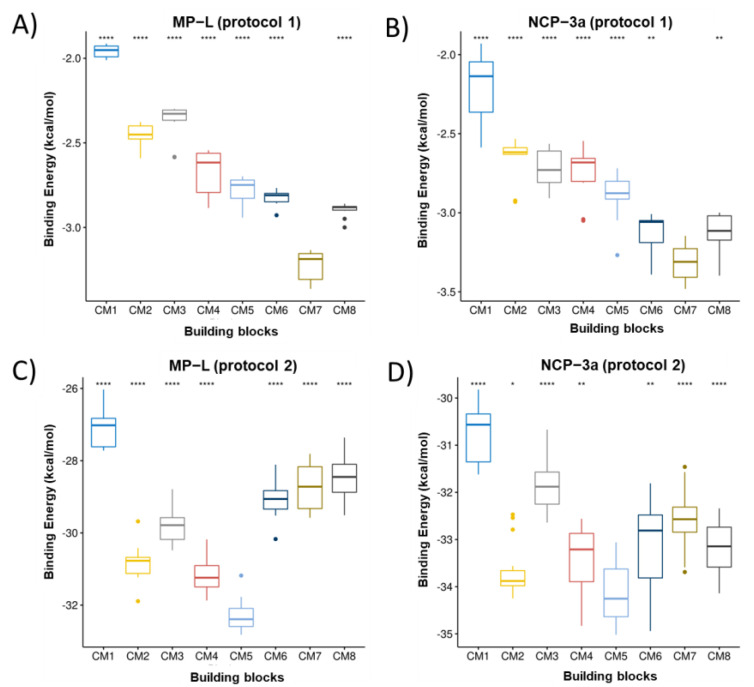
Ranking of CM. The boxplots of the cAMP-CMs binding energies calculated from the VINA scoring function are shown in (**A**,**B**) for the peptides MP-L and NCP-3a, respectively. The same as (**A**,**B**) but for the binding energy obtained from the glide scoring function (IFd) are shown in (**C**,**D**). * correspond to *p* < 0.05, ** correspond to *p* < 0.01 and **** correspond to *p* < 0.0001 of comparative statistical difference (using the *t*-test) with the lowest energy building block.

**Figure 4 pharmaceutics-15-00474-f004:**
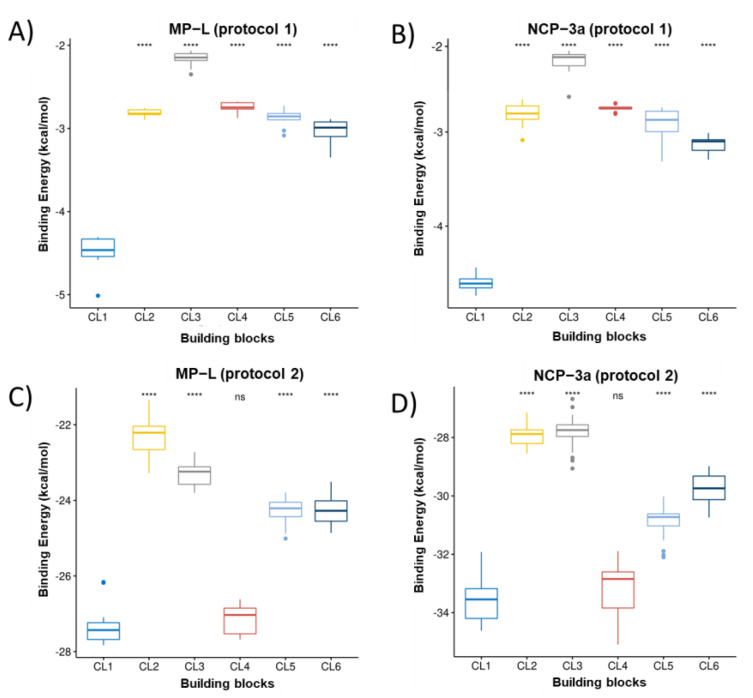
Ranking of CL. The boxplots of the cAMP-CLs binding energies calculated from the VINA scoring function are shown in (**A**,**B**) for the peptides MP-L and NCP-3a, respectively. The same as (**A**,**B**) but for the binding energy obtained from the glide scoring function (IFd) are shown in (**C**,**D**). **** correspond to *p* < 0.0001 of comparative statistical difference (using the *t*-test) with the lowest energy building block.

**Figure 5 pharmaceutics-15-00474-f005:**
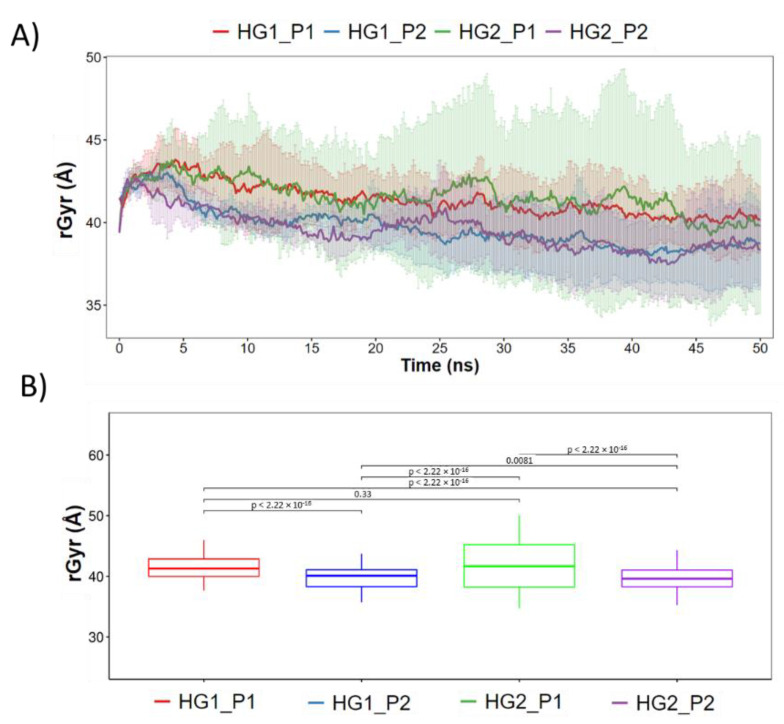
Radius of gyration (rGyr) of the hydrogels during the 50 ns of simulation. In (**A**) is shown the rGyr along the whole trajectory. (**B**) statistical analysis of the studied groups.

**Figure 6 pharmaceutics-15-00474-f006:**
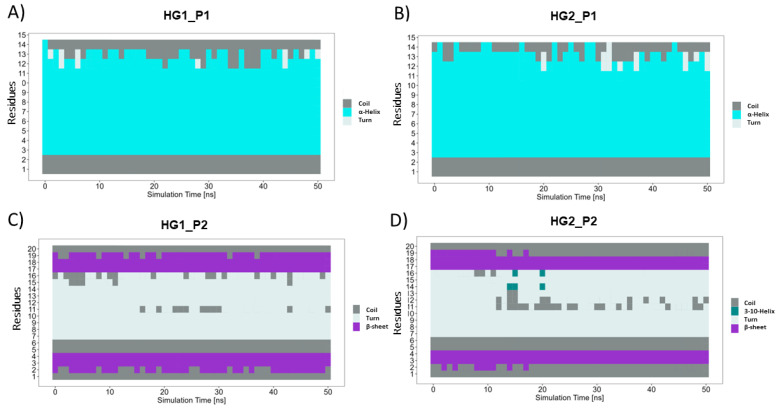
SSE of each peptide residue during 50 ns of simulation. P1 is in contact with HG1 and HG2 in (**A**,**B**), respectively. P2 in contact with HG1 and HG2 in (**C**,**D**), respectively. Each SSE is shown in a different color. The graph below summarizes the SSE composition for each trajectory time over the simulation.

**Figure 7 pharmaceutics-15-00474-f007:**
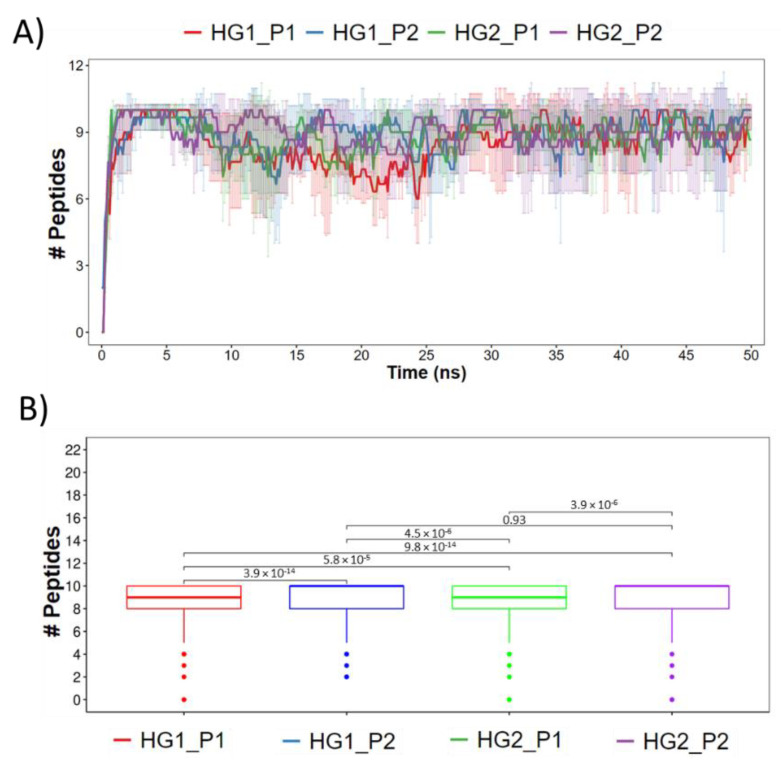
cAMPs that were less than 3 Å from the hydrogels during the entire simulation time. In (**A**) the count of cAMP molecules is shown during the entire simulation trajectory averaging between the 3 replicates (± standard deviation). (**B**) a statistical comparison of groups of cAMP counts.

**Figure 8 pharmaceutics-15-00474-f008:**
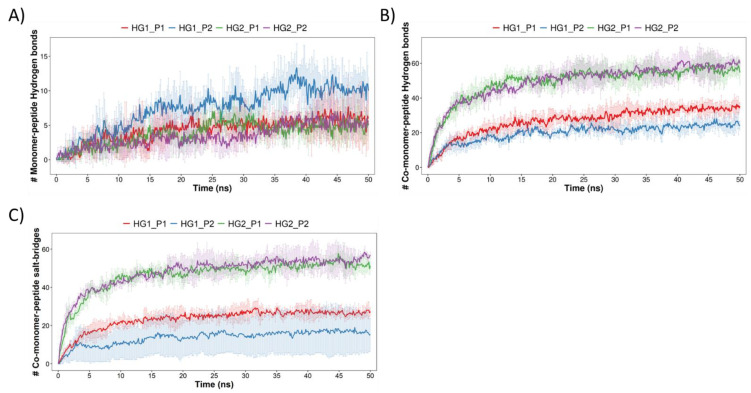
Intermolecular interactions formed between hydrogels and cAMPs during the 50 ns of simulation for MN-cAMP, CM-cAMP through HBs, (**A**,**B**), respectively. And for CM-cAMP through SBs in (**C**). The HBs were computed with a maximum distance of 2.8 Å and donor and acceptor angles of 120° and 90°, respectively. The maximum distance of SBs was 5.0 Å.

**Figure 9 pharmaceutics-15-00474-f009:**
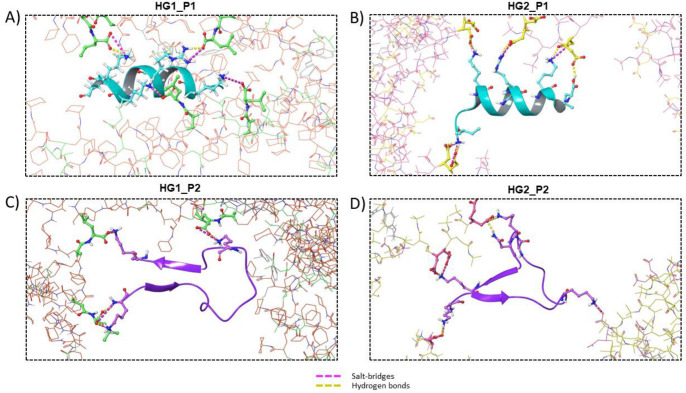
Interactions formed between hydrogels and cAMPs. In (**A**–**D**) are shown the four systems studied by MDs: HG1_P1, HG2_P1, HG1_P2, and HG2_P2, respectively. The alpha helix structure is shown in cyan, whereas the beta sheet structure is shown in purple.

**Figure 10 pharmaceutics-15-00474-f010:**
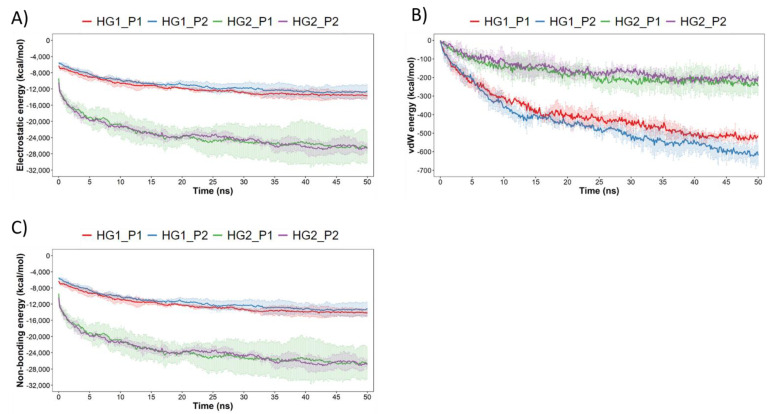
Interaction energies between hydrogels and peptides during the 50 ns of simulation. In (**A**–**C**) are shown the electrostatic, vdW and Non−bonding (electrostatic + vdW) energies computed along the whole trajectory in each system.

**Table 1 pharmaceutics-15-00474-t001:** Peptides properties.

	MP-L or P1	NCP-3a or P2
Source	Synthetic construct	Synthetic construct
Family	Derived from the peptide Mastoparan	Derived from CTX-1
Sequence	I N L K ILA R LA KK IL	K LIFIL S K T IPAG K N LF Y K I
Length	14 residues	20
Biological Activity	Anti-Gram positive, Anti-Gram negative, Antifungal	
Target Organism	***S. aureus*** (ATCC 25923) *S. pyogenes* (ATCC 19615) *L. ivanovii* (Li 4pVS2) ***P. aeruginosa*** (ATCC 27853) ***K. pneumoniae*** (ATCC 13883) ***A. baumannii*** (ATCC 19606) *C. albicans* (ATCC 90028) *C. parapsilosis* (ATCC 22019)	***S. aureus*** (ATCC 43300, ATCC 22953) *E. hirae* (ATCC 10541) *S. agalactiae* (ATCC 13813) *E. coli* (ATCC 25922) ***P. aeruginosa*** (ATCC 27853) ***A. baumannii*** (ornithological, cloacal) ***K. pneumoniae*** (herpetological, cloacal) *B. cepacia* (ATCC 17759) *M. catarrhalis* (ATCC 25238) *C. albicans* (ATCC 10231) *C. glabrata* (ATCC 90030) *M. pachydermatis* (DSMZ 6172)
Structure predicted	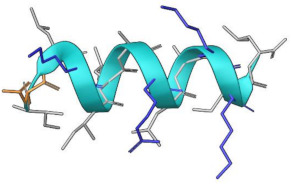	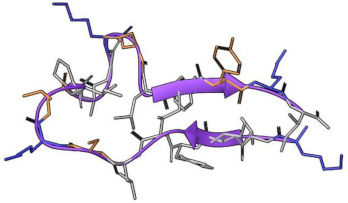

The non-polar, polar and positively charged amino acids are depicted in gray, orange and blue, respectively. The bacteria highlighted in bold belong to the ESKAPE pathogen group.

## Data Availability

Not applicable.

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
