# Peer review of "Rational Design of Hydrogels for Cationic Antimicrobial Peptide Delivery: A Molecular Modeling Approach"

_pharmaceutics, 2023, doi:10.3390/pharmaceutics15020474_

Round 1
Reviewer 1 Report
Dear Authors,
Authors focus on development of localized delivery systems of antimicrobial peptides using theoretical-computational methods,The topic addressed by the authors is relevant in the field. The originality aspects can be better highlighted. Authors proposed methods for designing hydrogels for antimicrobial delivery can improve significantly the time and accuracy of researchers in evaluating all possibilities when developing or testing similar systems . The methodology approach is well organized and controls are selected rationally and scientifically justified.
I suggest a minor revision:
2. The abstract can be improved. It should focus on current state of the antimicrobial delivery systems and accentuate the aim of the study.
3. Also, I recommend that in the conclusion section to be included some correlations between obtained results and literature findings.
Author Response
- The abstract can be improved. It should focus on current state of the antimicrobial delivery systems and accentuate the aim of the study.
Response: the abstract was improved according to the suggestions of the reviewer
- Also, I recommend that in the conclusion section to be included some correlations between obtained results and literature findings.
Response: the conclusion was improved according to the requirement of the reviewer. The changes are depicted as follows:
“The importance of hydrogels as drug delivery systems is unquestionable, their selection and optimization can be effectively achieved with theoretical methods [24], [34], as is reported by several studies such as the study of Changying et. al. [35], where through experimental validation they were able to confirm a computational pipeline like the reported here and thereby identify telodendrimer nanocarriers for specific drugs.”
Reviewer 2 Report
I recommend a few modification.
Double check the English by a nativ speker.
Please describe the abbreviations when you use them for the first time
Rephrase same part of paragraph:
" At a general level, the results show that the best candidates obtained coincide between the peptides studied, however, important differences are observed between the proposed strategies. In the monomer ranking, when protocol 1 was used, the best candidate for both peptides was MN3, and in the case of protocol 2, the best candidate for both peptides was MN4. For the co-monomer ranking, using protocol 1, the best candidate for both peptides was CM7, however, in the case of protocol 2, the best candidate for both peptides was CM5. Finally, in the cross-linkers ranking, for both strategies and considering the 2 peptides studied, the best candidate turned out to be CL1. It is important to mention that, when protocol 2 is used, CL4 also appears as a good candidate, showing no significant difference with CL1, however, the median score value is still lower in CL1. The main differences found between the building blocks when comparing both strategies may be due to the docking approach used in each protocol. As mentioned above, AutoDock uses a semi-flexible variant, in contrast IFd allows the blocks and peptide to be treated as flexible way at the same time. In practice, the binding energy values obtained with both protocols are not comparable because they are obtained from different scoring functions. Therefore, it was opted to select the best candidates in each protocol for the following steps. So, considering both strategies, the proposed hydrogel candidates were formed by the following blocks: MN3-CM7-CL1 (protocol 1) and MN4-CM5-CL1 (protocol 2). For next analysis these hydrogels (HG) will be named as HG1 (derived from protocol 1) and HG2 (derived from protocol 2)."
Figure 6 is not clear
I think that you need three short conclusions!
My recommendation is to focus on 3 short conclusion.
Author Response
I recommend a few modifications:
- Double check the English by a nativ speker.
Response: A native English speaker has reviewed the grammar of this article and the corrections.
- Please describe the abbreviations when you use them for the first time.
Response: The improvement regarding the abbreviations was performed
- Rephrase same part of paragraph:
" At a general level, the results show that the best candidates obtained coincide between the peptides studied, however, important differences are observed between the proposed strategies. In the monomer ranking, when protocol 1 was used, the best candidate for both peptides was MN3, and in the case of protocol 2, the best candidate for both peptides was MN4. For the co-monomer ranking, using protocol 1, the best candidate for both peptides was CM7, however, in the case of protocol 2, the best candidate for both peptides was CM5. Finally, in the cross-linkers ranking, for both strategies and considering the 2 peptides studied, the best candidate turned out to be CL1. It is important to mention that, when protocol 2 is used, CL4 also appears as a good candidate, showing no significant difference with CL1, however, the median score value is still lower in CL1. The main differences found between the building blocks when comparing both strategies may be due to the docking approach used in each protocol. As mentioned above, AutoDock uses a semi-flexible variant, in contrast IFd allows the blocks and peptide to be treated as flexible way at the same time. In practice, the binding energy values obtained with both protocols are not comparable because they are obtained from different scoring functions. Therefore, it was opted to select the best candidates in each protocol for the following steps. So, considering both strategies, the proposed hydrogel candidates were formed by the following blocks: MN3-CM7-CL1 (protocol 1) and MN4-CM5-CL1 (protocol 2). For next analysis these hydrogels (HG) will be named as HG1 (derived from protocol 1) and HG2 (derived from protocol 2)."
Response: the paragraph was rewritten.
- Figure 6 is not clear
Response: the description of figure 6 was improved.
- I think that you need three short conclusions!
Response: the conclusion was improved according to the requirement of the reviewer.
- My recommendation is to focus on 3 short conclusions.
Response: the conclusion was improved according to the requirement of the reviewer.
Reviewer 3 Report
It is a pity that the simulations were not supported by at least the simplest verification. They still remain mere conjecture, and not everything that comes into contact with a living organism is predictable.
Author Response
- It is a pity that the simulations were not supported by at least the simplest verification. They still remain mere conjecture, and not everything that comes into contact with a living organism is predictable.
Response: We thank the reviewer for his appreciation and comments. The main objective of this research article was to carry out a rational design accurately based on a set of theoretical-computational methods that allow specific prediction of hydrogel-cationic antimicrobial peptide complex to a possible synthesis of hydrogel with those building blocks. For this reason, carrying out assays with microorganisms is outside of the goals of the article.